# Industry 4.0 HUB: A Collaborative Knowledge Transfer Platform for Small and Medium-Sized Enterprises

**Alberto Cotrino *** , **Miguel A. Sebastián** and **Cristina González-Gaya**

Department of Construction and Manufacturing Engineering, Universidad Nacional de Educación a Distancia (UNED), C/Juan del Rosal 12, 28040 Madrid, Spain; msebastian@ind.uned.es (M.A.S.); cggaya@ind.uned.es (C.G.-G.)

* Correspondence: acotrino3@alumno.uned.es; Tel.: +49-175-815-6948

**Abstract:** Industry 4.0 brings opportunities for small- and medium-sized enterprises (SMEs), but SMEs are lacking Industry 4.0 knowledge, and this might result in a challenge to support SMEs' competitiveness and productivity. During recent years, the European Commission and some government initiatives have been fostering the transition toward Industry 4.0 for SMEs through the creation of *Digital Innovation Hubs*, the *Plattform Industrie 4.0*, and some other initiatives. Nonetheless, the authors consider that the lack of knowledge is still a risk toward Industry 4.0 transformation for SMEs. New ways to improve Industry 4.0 knowledge management and especially the knowledge transfer must be developed. When SMEs start the transition to Industry 4.0, first of all, they do not want to start from scratch, and secondly, it can be easy to get lost in the multitude of technologies and tools that are available in today's market. There is a gap in which to provide a collaborative Industry 4.0 knowledge transfer platform or hub designed for SMEs. Therefore, this research aims to enhance Industry 4.0 knowledge transfer through the development of a collaborative, web-based knowledge transfer Industry 4.0 platform. The outcome of this research is a developed platform that will be referred to as Industry 4.0 HUB.

**Keywords:** Industry 4.0; knowledge transfer; hub; platform

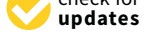



## 1. Introduction

The Fourth Industrial Revolution is profoundly changing the structures and processes within companies. Industry 4.0 is a one-time event that brings opportunities and challenges for large-sized enterprises, as well as for SMEs [1,2]. Big companies have deployed investment plans, roadmaps, and worldwide investigation groups to support the transition toward Industry 4.0. The close relationship between a large-sized enterprise and an SME requires both advancing and progressing at the same speed. Therefore, SMEs must adapt to Industry 4.0 developments proposed by large companies. To cope with this transformation and be able to adopt Industry 4.0 innovations, SMEs must demonstrate flexibility and adaptability. However, many SMEs still find it difficult to know which Industry 4.0 technologies to invest in [3–5].

Research and European initiatives in recent years proved that SMEs must be considered independently from large-sized enterprises regarding Industry 4.0 implementation, because they are less capable of handling the financial, technological, and staffing challenges than large companies [6–10]. The Fourth Industrial Revolution could end up being an obstacle to support firm competitiveness and productivity if the SMEs' peculiarities are not properly identified and supported. To gain a better insight into the characteristics of SMEs adopting Industry 4.0 technologies, the authors performed a systematic literature review to understand the challenges and problems that SMEs are facing during the Industry 4.0 transformation. Some authors refer to financial resources and technology awareness as the primary challenges that the SMEs are facing.

Other research points out high costs, a lack of information, complexity, and the abundance of technologies as the key challenges and problems. However, all of them agree on one challenge: a lack of knowledge [1–5,11,12]. SMEs have problems accessing Industry 4.0 knowledge due to the lack of knowledge management and transfer.

Knowledge management (KM) is a field of study that has existed for more than four decades [13]. It can be defined as the process of creating, sharing, using, and managing the knowledge and information of an organization [14,15]. KM has four primary activity areas or knowledge flows, as shown in Figure 1: knowledge creation, retention, transfer, and utilization [16,17].

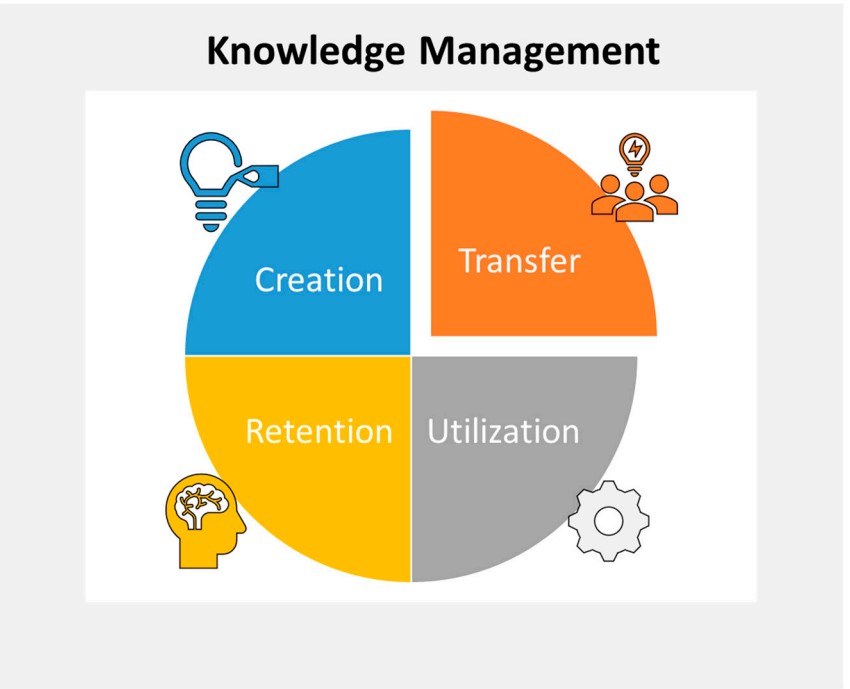

**Figure 1.** Knowledge management and the four knowledge flows.

This research focuses on Industry 4.0 knowledge transfer (KT) within SMEs, as this was the primary challenge identified during the literature review. KT, sometimes also known as knowledge sharing (KS), refers to sharing knowledge from one individual or group of individuals (e.g., an SME) to another individual or group of individuals. KT pursues to create, organize, and distribute knowledge between different users and seeks to ensure its availability to future users [15–18]. KT is a complex field because of the plurality of sources, including individuals, organizations, the internet, and books. Some aspects hindering successful KT are communication issues, a lack of networking and difficulty accessing the knowledge [19,20]. Therefore, the ability to transfer knowledge effectively in the networks of SMEs is of paramount importance for supporting the transition to Industry 4.0. Several studies suggest the development of online platforms to optimize KT [21,22]. These kinds of platforms promote a knowledge-sharing culture through discussion forums and other transfer tools so that SMEs contribute to the overall success.

The authors consider two aspects to be crucial for the success of SMEs: the improvement of the access to Industry 4.0 knowledge and Industry 4.0 KT. For that reason, this research aims to fulfil the following research objectives:

1.  Analyze the existing KT initiatives and platforms supporting the Industry 4.0 transition within SMEs;
2.  Propose a new collaborative online platform to improve Industry 4.0 KT between SMEs.

## 2. Materials and Methods

In this work, we applied a research approach in two phases:

- The first phase was a literature review conducted in 2019 and 2020 to evaluate current Industry 4.0 knowledge transfer initiatives and platforms. This will be described in Section 2.1.
- For the second phase, by the end of 2020, the collaborative online platform Industry 4.0 HUB for knowledge transfer for SMEs was developed. This phase will be explained in Sections 2.2 and 2.3.

### 2.1. Literature Review

The authors performed a systematic literature review using several databases such as the Web of Science, Google Scholar, and Scopus to identify existing Industry 4.0 KT initiatives and platforms.

The foundation of *Plattform Industrie 4.0* by several German associations with more than 6000 member companies in April 2013 was the first attempt to create a platform supporting industry KT and the further development of the recent trend in manufacturing technologies, termed as Industry 4.0. The platform, officially announced and kicked off at the Hannover Fair in 2013, seeks to promote the development of Industry 4.0 in Germany and thus strengthen the competitiveness of Germany as a production location [23,24]. In this context, the excellent Industry 4.0 map is worth highlighting due to its transformation of knowledge related to successful and sustained Industry 4.0 test cases and support of other SMEs accessing Industry 4.0 technologies by providing already proven solutions and saving the initial trial and error phase. In addition, the SME Transfer Network submodule organizes meetings and workshops to promote KT within SMEs.

Using the benefits of the digital transformation, the *Digital Innovation Hub* (DIH) from the European Commission is another excellent initiative to further promote Industry 4.0 KT within SMEs. It intends to build a European network of DIHs that are one-stop shops to help companies improve their processes, products, and services by using digital technologies. DIHs provide access to technical expertise and experimentation by acting as knowledge transfer platforms [25–27]. To support this initiative, the European Commission has proposed the creation of the first-ever Digital Europe Programme, which will invest EUR 9.2 billion to align the next long-term EU budget (2021–2027) with increasing digital challenges [27–29]. The current numbers show the astonishing progress of the DIHs, as shown in Figure 2. By May 2021, there were 350 fully operational DIHs in Europe. Countries like Spain and Italy already have more than 50 DIHs fully operational, while other countries like Germany and France have more than 20 DIHs in preparation that will open their doors in the following months [30]. The Digital Europe Programme will foster the creation of new DIHs even further.

During recent years, a new trend has been observed: some DIHs are getting connected and creating a cluster to expand their area of influence to a national level and to lead the development not only of a region, but also of a nation or even a pan-European area [31]. One example of this development is the cluster de:hub, which comprises twelve DIHs creating one digital ecosystem [29]. Furthermore, a relevant part of the DIHs is the project DIHNET.EU, which aims to create a sustainable pan-European network of networks to interconnect the regional DIHs and promote networking and collaboration [32]. This project might help to create a great online community to foster interaction among hubs, information exchange, and peer learning.

To conclude, and to highlight the relevance of the abovementioned platforms, the interest and research in recent years on the DIH and *Plattform Industrie 4.0* is growing, as shown in Figure 3. Moreover, the number of searches for DIH in Google in the last four years shows popularity peaks during the last two years, which emphasizes the need for Industry 4.0 knowledge transfer platforms, as shown in Figure 4.

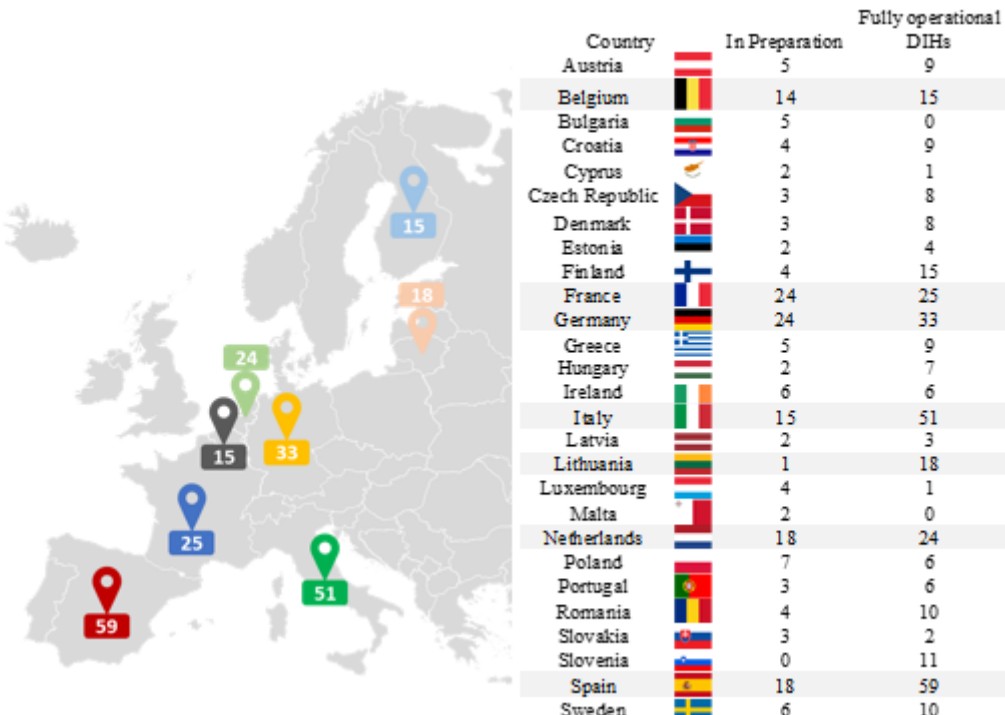

| Country | | In Preparation | Fully operational DIHs |
|---|---|---|---|
| Austria | | 5 | 9 |
| Belgium | | 14 | 15 |
| Bulgaria | | 5 | 0 |
| Croatia | | 4 | 9 |
| Cyprus | | 2 | 1 |
| Czech Republic | | 3 | 8 |
| Denmark | | 3 | 8 |
| Estonia | | 2 | 4 |
| Finland | | 4 | 15 |
| France | | 24 | 25 |
| Germany | | 24 | 33 |
| Greece | | 5 | 9 |
| Hungary | | 2 | 7 |
| Ireland | | 6 | 6 |
| Italy | | 15 | 51 |
| Latvia | | 2 | 3 |
| Lithuania | | 1 | 18 |
| Luxembourg | | 4 | 1 |
| Malta | | 2 | 0 |
| Netherlands | | 18 | 24 |
| Poland | | 7 | 6 |
| Portugal | | 3 | 6 |
| Romania | | 4 | 10 |
| Slovakia | | 3 | 2 |
| Slovenia | | 0 | 11 |
| Spain | | 18 | 59 |
| Sweden | | 6 | 10 |

**Figure 2.** Top 7 EU countries by number of fully operational DIHs.

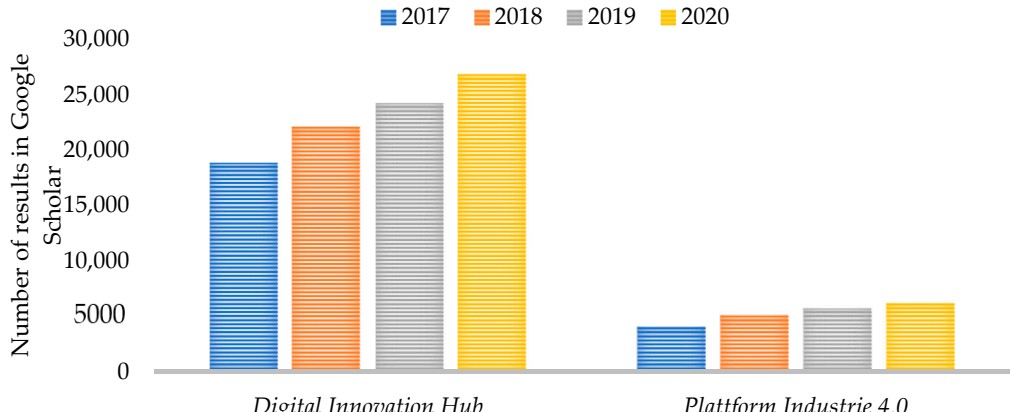

**Figure 3.** Development during the last four years of the number of results in Google Scholar related to DIH and *Plattform Industrie 4.0*.

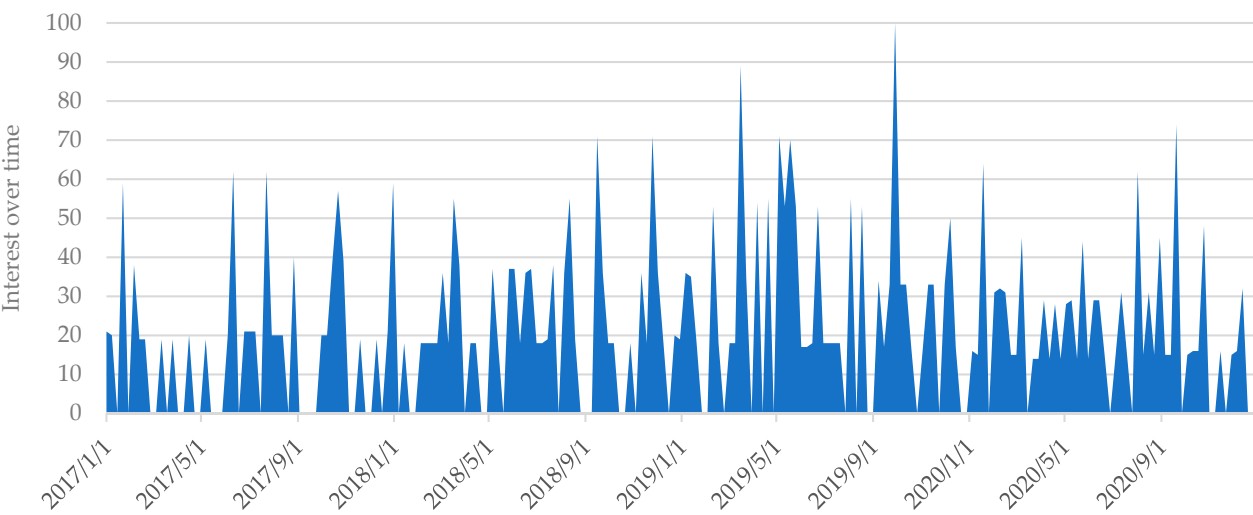

**Figure 4.** Interest over the last four years in DIH, generated via Google Trends.

### 2.2. Industry 4.0 HUB

In summation, SMEs are being supported by European initiatives in order to improve KT in regard to Industry 4.0. Moreover, there are even Industry 4.0 KT platforms, clusters, and hubs for SMEs, like the DIHs or *Plattform Industrie 4.0*, which will increase their influence and support SMEs in the following years through European investment plans.

In the case of DIHs, the European Commission is creating a complex, regional, multi-layered, and heterogeneous innovation ecosystem that will promote its expansion in the following years. Some studies have suggested the importance of addressing Industry 4.0 at a regional level to foster innovation locally [33–35]. However, given the number, the different focus, and the specialization of DIHs at the regional, national, and EU levels, new strategies to coordinate, control, and optimize communication through those levels are necessary. DIHs are regionally based, supporting the local industry. Nonetheless, collaboration and networking between the DIHs are essential to ensure that knowledge related to the best practices can be transferred.

Industry 4.0 technologies are evolving fast, and knowledge must be collected in real time. Moreover, the inclusion of new technologies, and especially the advent of Industry 4.0, are facilitating collaboration while at the same time accelerating the development of innovation outcomes and setting new challenges for SMEs [25,28,31,36]. New KT strategies must be developed considering the dynamics resulting from the implementation of Industry 4.0. Consequently, promoting a collaborative atmosphere by boosting the synergies between SMEs and promoting networking through the different levels previously mentioned will enhance a culture of innovation.

When SMEs start the transition to Industry 4.0, first of all, they do not want to start from scratch, and secondly, it can be easy to get lost in the multitude of technologies and tools that are available in today's market. There is a lack of an essential Industry 4.0 collaborative knowledge platform for every SME at every level, as shown in Figure 5. There is a gap for providing a collaborative web application Industry 4.0 KT hub without borders and without the level of granularity that initiatives like DIHs have created (European countries, European regions, and specialized DIHs).

The KT of Industry 4.0 technologies and projects, as well as the networking of SMEs, can be facilitated via the creation of a collaborative hub. This research aims to fill this gap and has its focus on developing a collaborative, web-based KT hub for the implementation of Industry 4.0 components in the supply chains of SMEs.

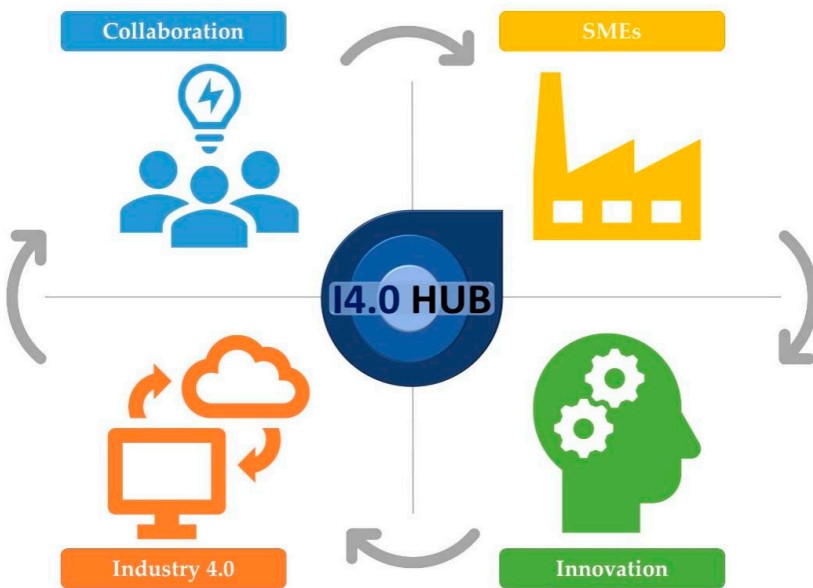

**Figure 5.** Industry 4.0 knowledge transfer for SMEs.

*2.3. Development of the Industry 4.0 HUB*

The web-based KT hub developed during this research is called Industry 4.0 HUB. It is a full software stack platform (frontend, backend, and database). There are several alternatives of web application software stack solutions that might be selected based on different decision criteria for each project, including the project size, scalability, maintainability, and security [37]. To support the decision-making, recent results from the Developer Survey 2020 from Stack Overflow were analyzed. Nearly 65,000 developers from over 180 countries participated in the survey, and the results show that JavaScript, HTML or CSS, and SQL are the top three programming, scripting, and markup languages used [38]. As this project is still in its early stages, the authors have selected the following alternatives based on their experience and the current usage between web developers. Other than that, other criteria were not taken into consideration. For the frontend development, the following programming languages were used:

- HTML 5: the latest version of the markup language for web pages;
- CSS 3: a style sheet language used for describing the presentation of a document written in a markup language such as HTML;
- Bootstrap 4: HTML, CSS, and JavaScript framework for developing responsive web pages.

The backend development was performed using the following programming languages:

- PHP: a server scripting language and a powerful tool for making dynamic and interactive web pages;
- JavaScript: high-level and multi-paradigm programming language.

Finally, MySQL was used for the creation of the database. Moreover, several libraries, such as the JavaScript library jQuery, were used to simplify the programming and expand the capabilities of the Industry 4.0 HUB. The full software stack architecture is shown in Figure 6.

The source code editor used for the development was Visual Studio Code. The first phase of the development was performed locally on the laptops of the authors on a local test server without interaction with a live server. Once the platform was running in a stable state and the first dry runs were completed, it was uploaded to a web server, and the Industry 4.0 HUB has been live since January 2021 from the following link: i40-hub.com.

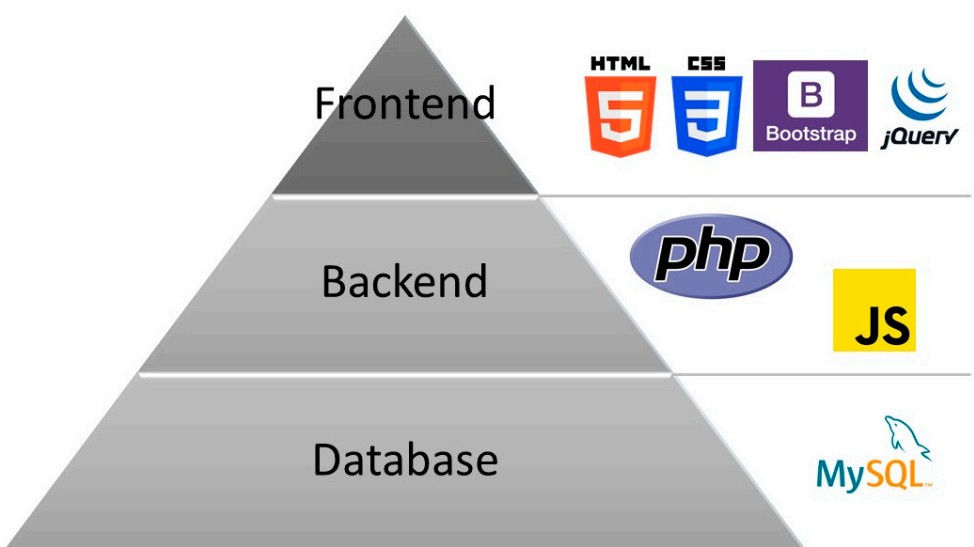

**Figure 6.** Frontend, backend, and database development.

## 3. Results

The Industry 4.0 HUB is a website that aims to support SMEs in selecting the appropriate Industry 4.0 technologies and projects. The website intends to work as an SME collaborative network for KT, where SMEs can access the latest knowledge, expertise, and technology to access Industry 4.0. The Industry 4.0 HUB wants to eliminate impediments to accessing Industry 4.0 knowledge; it wants to offer open access to Industry 4.0 knowledge.

This platform does not pretend to gather single Industry 4.0 technologies, but instead collects Industry 4.0 use cases, real solutions, and projects in SMEs. The Industry 4.0 HUB is based on three pillars: the SMEs accessing the platform, the website offering the front end capabilities, and a database, as the backend, where Industry 4.0 use cases are stored (see Figure 7). The Industry 4.0 HUB is designed with a modular architecture that guarantees great advantages in terms of flexibility and extensibility. The Industry 4.0 HUB offers four modules, which will be explained in the following subsections.

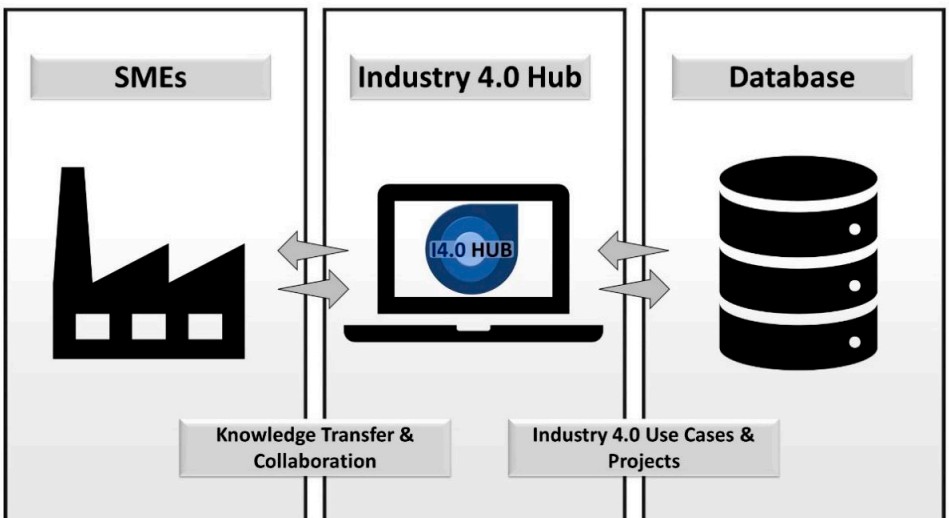

**Figure 7.** Industry 4.0 HUB architecture.

### 3.1. The HUB

This module is the core of the KT. The HUB enables the SMEs to access the Industry 4.0 database and gain the information and awareness necessary to implement Industry

4.0 use cases. Each use case is presented from a practical viewpoint, with a detailed dataset including the fields presented in Table 1.

**Table 1.** Dataset of the Industry 4.0 use cases.

| Field | Description |
|---|---|
| Name of the Industry 4.0 Project | Unique identification of the Industry 4.0 use case, which is used as a key for the SQL database. |
| Industry 4.0 Technology | The literature about technologies related to Industry 4.0 is extensive [39–42]. Considering the large number of technologies related to Industry 4.0, the authors have selected six of these technologies to classify the use cases: Internet of Things, machine learning, automation, big data, cloud computing, augmented reality, and other. |
| Description | Short introduction of the Industry 4.0 solution, including the use case of the project and the improvement achieved. With this information, other SMEs can gain a quick understanding of and easily and quickly comprehend the key usage of the solution and implement it in their organizations. |
| Budget | Several research works have highlighted that SMEs do not have the economic resources to implement Industry 4.0 technologies [5,8]. For the Industry 4.0 HUB, the authors included this category so that microenterprises and small enterprises with turnover ranges between EUR 0.5 million and EUR 10 million can carefully select the solution based on the costs and the investment. |
| Picture | This allows the user to upload a picture of the use case to support rapid visual recognition. |
| Additional Information (Link) | This is a field to include links to provide additional information, such as the website of the vendor of the technologies used. |

Some examples of Industry 4.0 use cases in SMEs related to the Internet of Things presented using the abovementioned dataset are shown in Figure 8.

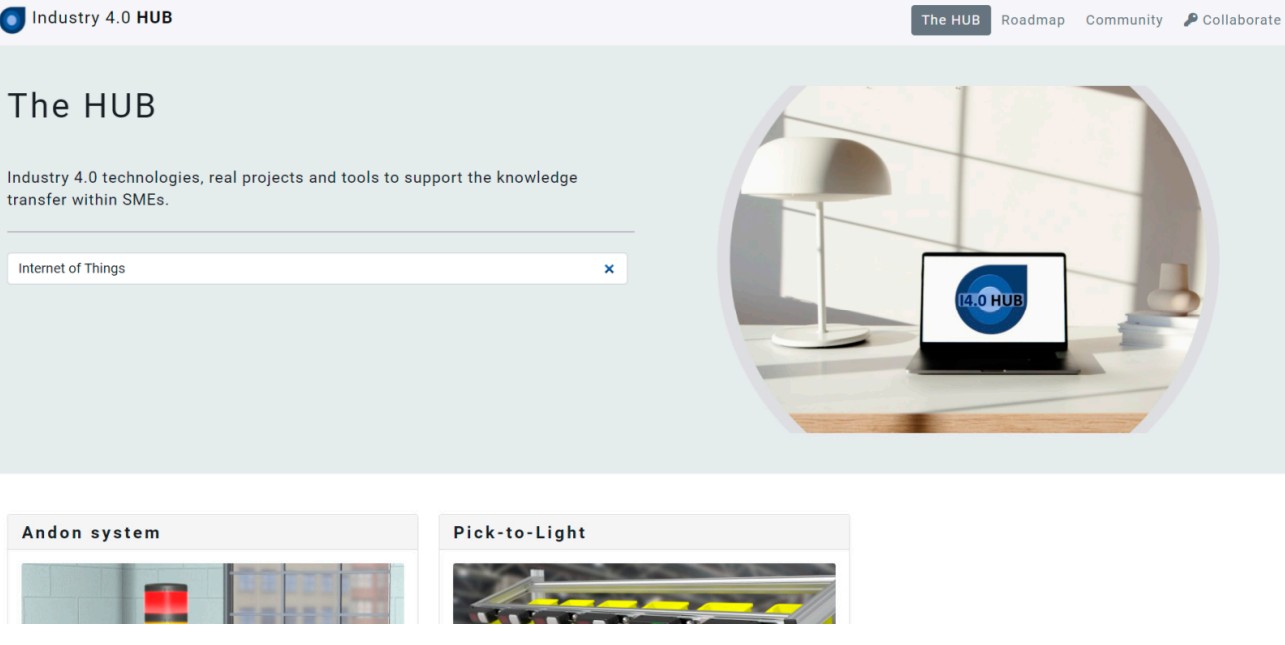

**Figure 8.** Industry 4.0 HUB desktop user interface.

## 3.2. Roadmap

The HUB enables Industry 4.0 KT and allows SMEs to gain the Industry 4.0 practical knowledge required to implement the projects in their facilities. However, the implementation could be tortuous sometimes, and therefore, the authors provide in the Industry 4.0 HUB an additional module to support the SMEs: Roadmap.

Roadmap aims to facilitate decision-making and access to Industry 4.0 technologies by the production areas of the SMEs. The six-step roadmap presented is based on the roadmap for SMEs published in November 2020 by the authors [8]. The main differences between the roadmap proposed within this paper and the roadmaps presented in previous research are its focus on SMEs and the practical viewpoint.

### 3.3. Community

Following the networking trend being observed during recent years in other platforms like DIH, the Industry 4.0 HUB proposes a module called Community. It aims to be a forum where SMEs can share experiences on Industry 4.0 implementation and where collaboration can appear.

### 3.4. Collaborate

This module allows SMEs to contribute to the Industry 4.0 HUB, introducing use cases, real projects, and implementations of SMEs in the database. Every DIH has a different focus and specialization on a regional level, which makes access to the knowledge for SMEs from other regions or nations within the European Union and outside of it difficult. On the other hand, *Plattform Industrie 4.0* focuses mainly on the use cases in Germany. Both platforms have already defined structures and procedures. Single users or SMEs are not able to collaborate; they cannot transfer Industry 4.0 knowledge with other SMEs. The Industry 4.0 HUB aims to create a platform without regional or national barriers and where everyone has the right to share Industry 4.0 knowledge and experience.

## 4. Discussion and Conclusions

The result of this research is the Industry 4.0 HUB platform, which has been completely developed and online since January 2021 at the following link: i40-hub.com. This research presents the qualitative advantages of using the collaborative KT platform Industry 4.0 HUB:

- The Industry 4.0 HUB creates a single source for Industry 4.0 KT for SMEs. Every SME willing to implement Industry 4.0 technologies does not need to seek hundreds of papers, books, and platforms as everything is available in just one website, which covers the lack of knowledge highlighted in Section 1. Additionally, the resources needed to start the Fourth Industrial Revolution are minimized.
- The HUB provides real-time SME access to an immense collection of Industry 4.0 use cases.
- The Collaborate and Networking modules allow SMEs to enter use cases and share Industry 4.0 knowledge and resources, creating a collaborative network.
- The Roadmap module helps SMEs enter the new era of manufacturing using a simple process that does not require complex expertise or big teams.

Industry 4.0 KT must be simplified for the SMEs, especially for the microenterprises, and therefore, a single source platform like the Industry 4.0 HUB could provide the solution to the deficiency identified. As was previously mentioned, DIHs and *Plattform Industrie 4.0* are great initiatives and key enablers for the transition toward Industry 4.0 and digital transformation for European SMEs. The European network of DIHs will help companies to improve their processes, products, and services using digital technologies, and they will provide access to technical expertise and experimentation, but the authors wanted to propose a different way to improve the Industry 4.0 KT between SMEs. Table 2 highlights some of the differences between the Industry 4.0 HUB, the *Plattform Industrie 4.0*, and the DIHs. In conclusion, the Industry 4.0 HUB supports the transition toward Industry 4.0 for SMEs by using a different approach than the one used in *Plattform Industrie 4.0* and the DIHs. The Industry 4.0 HUB aims to eliminate unnecessary barriers and to simplify Industry 4.0 KT.

However, this study has some limitations which may be addressed through future studies. The first limitation is that further analysis of the quantitative results for the Industry 4.0 HUB is still required. A representative number of SMEs need to start using

the platform to start gathering data and scaling the solution to adapt it to each SME's needs. The second limitation is the development of the Industry 4.0 HUB website, which requires further improvement and bug fixing, as well as the creation of new capabilities for SMEs. Another possibility that should be studied is the usage of a different programming framework, such as a content management system (CMS) like Drupal, which could simplify the scalability and maintainability of the platform.

**Table 2.** Comparison between Industry 4.0 HUB, DIH, and *Plattform Industrie 4.0*.

| Platform | Scalability | Resources | Networking | Technical Expertise | Specialization |
|----------|-------------|-----------|------------|---------------------|----------------|
| DIH | ✗ | ✗ | ✓ | ✓ | ✓ |
| *Plattform Industrie 4.0* | ✓ | ✗ | ✗ | ✓ | ✓ |
| Industry 4.0 HUB | ✓ | ✓ | ✓ | ✗ | ✗ |

**Author Contributions:** A.C.: conceptualization, methodology, investigation, writing—original draft preparation, and visualization; M.A.S.: conceptualization, methodology, investigation, and writing—review and supervision; C.G.-G.: conceptualization, methodology, investigation, and writing—review and supervision. All authors have read and agreed to the published version of the manuscript.

**Funding:** This research was funded by the Annual Grants Call of the International Doctorate School of the Spanish National Distance-Learning University (EIDUNED).

**Institutional Review Board Statement:** Not applicable.

**Informed Consent Statement:** Not applicable.

**Data Availability Statement:** The data presented in this study is available within this article.

**Acknowledgments:** This paper has been produced within the scope of the doctoral activities carried out by the lead author at the International Doctorate School of the Spanish National Distance-Learning University (EIDUNED_ Escuela Internacional de Doctorado de la Universidad Nacional de Educación a Distancia). The authors are grateful for the support provided by this institution.

**Conflicts of Interest:** The authors declare no conflict of interest.

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
