# Peer review of "Industry 4.0 HUB: A Collaborative Knowledge Transfer Platform for Small and Medium-Sized Enterprises"

_applsci, doi:10.3390/app11125548_

Round 1

Reviewer 1 Report

The article was easy to read, and presents a good and interesting topic.
I only have a couple of remarks, which can/should be considered:
 - In the section titled "Materials and methods" - The section names are not displayed properly in the description. ("Section 0").  
 - I believe that using google trend online service, - where the number of searches in the last couple of years are shown -, could highlight the need for HUBs like the developed one.

Reviewer 2 Report

It is unclear to me whether the Industry 4.0 HUB has been built or not. Please include a clarifying statement in the abstract, conclusion and discussion sections.

Some typos and word choice changes are marked on the document. 

Line 39 needs a word change.
